# Challenges and facilitators for health professionals providing primary healthcare for refugees and asylum seekers in high-income countries: a systematic review and thematic synthesis of qualitative research

Luke Robertshaw,[1] Surindar Dhesi,[2] Laura L Jones[1]

► Prepublication history and additional material are available. To view these files, please visit the journal online (http://dx.doi.org/10.1136/bmjopen-2017-015981).

[1]Institute for Applied Health Research, University of Birmingham, Birmingham, UK
[2]Department of Earth and Environmental Sciences, School of Geography, University of Birmingham, Birmingham, UK

**Correspondence to**
Dr. Laura L Jones;
L.L.Jones@bham.ac.uk

## ABSTRACT

**Objectives** To thematically synthesise primary qualitative studies that explore challenges and facilitators for health professionals providing primary healthcare for refugees and asylum seekers in high-income countries.

**Design** Systematic review and qualitative thematic synthesis.

**Methods** Searches of MEDLINE, EMBASE, PsycINFO, CINAHL and Web of Science. Search terms were combined for qualitative research, primary healthcare professionals, refugees and asylum seekers, and were supplemented by searches of reference lists and citations. Study selection was conducted by two researchers using prespecified selection criteria. Data extraction and quality assessment using the Critical Appraisal Skills Programme tool was conducted by the first author. A thematic synthesis was undertaken to develop descriptive themes and analytical constructs.

**Results** Twenty-six articles reporting on 21 studies and involving 357 participants were included. Eleven descriptive themes were interpreted, embedded within three analytical constructs: healthcare encounter (trusting relationship, communication, cultural understanding, health and social conditions, time); healthcare system (training and guidance, professional support, connecting with other services, organisation, resources and capacity); asylum and resettlement. Challenges and facilitators were described within these themes.

**Conclusions** A range of challenges and facilitators have been identified for health professionals providing primary healthcare for refugees and asylum seekers that are experienced in the dimensions of the healthcare encounter, the healthcare system and wider asylum and resettlement situation. Comprehensive understanding of these challenges and facilitators is important to shape policy, improve the quality of services and provide more equitable health services for this vulnerable group.

## Strengths and limitations of this study

► This is the first review to systematically identify and synthesise qualitative research exploring challenges and facilitators for health professionals providing primary healthcare for refugees and asylum seekers.
► Thematic synthesis of studies from a range of countries and primary healthcare settings allows identification of common, generalisable themes with potential to influence policy and practice.
► The review was limited to English language studies, which may have led to over-representation of studies conducted in English-speaking high-income countries.
► The review was limited to core, clinical health professionals: doctors nurses and midwives.

and countries due to violence or threats of violence. Other nations may provide refuge for those seeking a safe haven, and in 1950, the Office of the United Nations High Commissioner for Refugees (UNHCR) was established to provide international leadership and coordination for the protection of refugees and promotion of their well-being.[1] The UNHCR convention defines refugees as persons who have a 'well-founded fear of being persecuted for reasons of race, religion, nationality, membership of a particular social group or political opinion, is outside the country of his nationality and is unable or, owing to such fear, is unwilling to avail himself of the protection of that country; or who, not having a nationality and being outside the country of his former habitual residence as a result of such events, is unable or, owing to such fear, is unwilling to return to it'.[2] Those in the application process to be granted refugee status are referred to

## INTRODUCTION

Throughout human history, countless people have been forced to flee from their homes

as 'asylum seekers'. By the end of 2015, there were an estimated 65.3 million forcibly displaced people worldwide, including 40.8 million internally displaced people, 21.3 million refugees and 3.2 million asylum seekers.[3]

Refugees and asylum seekers are a vulnerable group with significant and complex health needs.[4] A survey by the UK Border Agency in 2010 showed refugees to be in poorer health than the general population.[5] As most refugees and asylum seekers originate from low-income and middle-income countries, there are, accordingly, higher prevalences of pre-existing infectious diseases such as hepatitis B, tuberculosis and HIV compared with host populations.[6] The risk of contracting infectious diseases may be increased by poor hygiene conditions during flight from conflict, coupled with insufficient vaccine coverage.[7] Studies have also highlighted the sexual and reproductive health needs of this group,[8] with high levels of sexual gender-based violence (SGBV) being reported along with limited access to contraception.[8 9] Refugees and asylum seekers also suffer from non-communicable diseases such as hypertension, musculoskeletal disease, chronic respiratory disease and diabetes, which may be undermanaged and exacerbated when they are forced to flee their countries.[10]

A further concern for refugee and asylum seeker populations is their mental health. Violence experienced in countries of origin, including war, sexual abuse and torture are reported, which may lead to psychological and physical trauma.[11] These premigration traumas are compounded by postmigration stressors such as loss of social networks, shifting societal roles and cross-cultural stress while integrating into countries of settlement.[12] Fazel *et al*[13] estimated that 9% of adult refugees may suffer with post-traumatic stress disorder, which is approximately 10 times estimates in an age-matched American population.[13]

Primary healthcare teams are on the frontline of healthcare provision for refugees and asylum seekers that arrive in high-income countries.[14] These teams may include a variety of professional backgrounds, clinical and non-clinical, but typically include a core of general practitioners, community-based nurses and midwives.[15 16] These health professionals face significant challenges when caring for refugees and asylum seekers.[17–19] They must address their complex health and social needs, often in cross-cultural interactions, and operate within health systems that may not be structurally configured or politically favourable towards this group.[17–20] These challenges impact on their ability to provide the same quality of care as the general population, leading to healthcare inequalities.[20 21]

Experiences of health professionals caring for refugees and asylum seekers in high-income countries have been investigated through a range of qualitative research studies conducted across several countries and primary healthcare settings. A recent systematic review by Suphanchaimat *et al*[22] synthesised challenges providing healthcare services to migrants from a provider perspective. The review included a minority of studies that had

refugees and asylum seekers as service users, focused purely on challenges of healthcare provision, and adopted a limited, purposive search strategy. To our knowledge, this present review is the first to synthesise experiences of health provision for migrants defined specifically as refugees and asylum seekers; synthesise both challenges and facilitators for health professionals and adopt a systematic approach to identification of qualitative research. Therefore, this review aims to systematically identify and thematically synthesise challenges and facilitators experienced by health professionals that provide primary healthcare for refugees and asylum seekers in high-income countries.

## METHODS

This systematic review sought qualitative research studies as they are the appropriate design for understanding perceptions and experiences of healthcare provision.[23 24] Systematic identification and synthesis of these studies may consolidate the current evidence-base, increase the breadth and depth of understanding and provide more generalisable conclusions than individual primary studies.[25 26]

This review was guided by established methodology for systematic review and thematic synthesis of qualitative research, outlined by Thomas and Harden.[27] Thematic synthesis of data, applied in this methodology, is suited to development of recommendations for practice and policy and provides a transparent link between conclusions and the primary studies synthesised.[27 28] Reporting of this review has been guided by Enhancing Transparency of Reporting the Synthesis of Qualitative Research (ENTREQ) framework.[29]

### Search strategy

The following databases were searched from inception until week 3 of March 2016: MEDLINE, EMBASE, PsycINFO, CINAHL and Web of Science. The search strategy was based on the Sample, Phenomenon of interest, Design, Evaluation, Research type (SPIDER) tool, which has been developed as an alternative to Population, Intervention, Comparison, Outcome (PICO) to optimise identification of qualitative studies for evidence syntheses.[30] Search terms were combined for primary health professionals/healthcare, refugees and asylum seekers and qualitative research. No language or date limits were applied. The full detailed search strategy is documented in online supplementary file 1. Further hand-searches were conducted based on included studies' reference lists and citations (in Google Scholar).

After removal of duplicates, titles and abstracts were screened by one researcher (LR), excluding articles that clearly did not meet the inclusion criteria. Full-texts of remaining articles were obtained and assessed by two independent researchers, according to prespecified study selection criteria (detailed below). Disagreements were resolved via discussion.

## Box 1 Definitions of challenge and facilitator

Challenge: a factor that inhibits, obstructs or creates difficulties for health professionals when providing primary healthcare.
Facilitator: a factor that promotes, enables or assists health professionals when providing primary healthcare.

### Selection criteria

This review included peer-reviewed, qualitative primary research studies that met the following criteria: English language; explored challenges or facilitators (defined in box 1) for health professionals providing primary healthcare to refugees and asylum seekers (including forced migrants, involuntary migrants or refugee claimants); and were conducted in a high-income country as defined by the World Bank country classification 2015.[31] Studies were limited to those from high-income countries because of the authors' interest in developing recommendations for policy and practice applicable to advanced primary healthcare systems.

Mixed methods studies were included if the qualitative element's methods and results could be isolated for synthesis. As definitions of health professionals in primary healthcare teams are diverse,[16] this review was limited to articles that interviewed core clinical healthcare professionals including: general practitioners, nurses, pharmacists and midwives working in primary healthcare settings. Articles were excluded if they were not based on peer-reviewed primary qualitative studies (ie, reviews, case studies, reports, opinion pieces) or were conducted in a secondary care setting. Articles that had referred to service users as 'migrants' or 'immigrants' were excluded, as these terms have a broader meaning including economic migrants, students and family unification.[32] Those that referred to 'illegal immigrants' or 'undocumented migrants' were also excluded as they are known to have unique characteristics (eg, ineligible for free healthcare) that would not be typical of refugees and asylum seekers.[33] Articles interviewing mental health professionals were excluded as this clinical area has specific characteristics. Studies that contained a mixture of eligible and ineligible participants were only included if data for eligible participants could be isolated for synthesis. Studies were also excluded if the full-text articles could not be obtained through institutional access or from requests sent to authors through Research Gate. The full inclusion and exclusion criteria applied in this review are documented in online supplementary file 2.

### Data extraction

Study characteristics were extracted by one author (LR) using a data extraction proforma. Characteristics included aims, setting, participants, methodology, results and recommendations/applications. Findings (results) and discussion sections from included articles were imported into NVivo V.11 software (NVivo qualitative data analysis Software; QSR International, V.11, 2016) for analysis.

### Assessment of quality

Included articles were assessed by one author (LR) using the Critical Appraisal Skills Programme (CASP) tool for appraisal of qualitative research.[34] Articles were not excluded from the synthesis or given weighting based on this assessment, as there is currently no accepted method for this in syntheses of qualitative research.[35] All articles were included irrespective of their reporting quality given that they contributed to the conceptual richness of the synthesis. Where articles used mixed methods, only the qualitative element was appraised.

### Data synthesis

A thematic synthesis was conducted broadly following the methodology outlined by Thomas and Harden.[27] An article, considered data-rich (containing numerous challenges and facilitators), was selected as an index-article and uploaded into NVivo V.11 software. The findings (results) and discussion sections were coded inductively within the two categories of 'challenges' and 'facilitators', as defined by the review question. This approach of inductive coding within a priori categories follows established methodology seen in similar qualitative syntheses.[36] Primary quotations, author's commentary and author's interpretations were coded. Sections were only coded if they contained challenges or facilitators (box 1), and referred to the health professionals defined for this review. Following the index-article, subsequent articles were coded using the same method in approximate order of descending data-richness. Concepts in each article were coded to iteratively develop and refine a codebook, with each article having an ability to contribute new codes. Once all articles had been coded, the finalised codebook was applied across all articles. The final codebook was analysed to inform descriptive themes closely resembling the prevailing concepts across primary studies. These themes were discussed and agreed within the research team. An analytical model was then developed to create higher-order constructs within which descriptive themes were located.

## RESULTS

### Systematic search and selection

Systematic database searches identified 5970 articles. A further 16 articles were identified through hand-searching of reference lists and citations. After removal of duplicates, 3571 articles remained. A total of 3493 articles were excluded based on the title and abstract. Full-texts of the remaining 78 articles were sought for detailed assessment against the inclusion criteria. Nine of these articles could not be obtained. In addition, due to resource limitations, four non-English language studies were unable to be translated and assessed against the selection criteria. After reviewing the 65 available full-text papers and applying the full selection criteria, 26 articles were included in the thematic synthesis (figure 1).

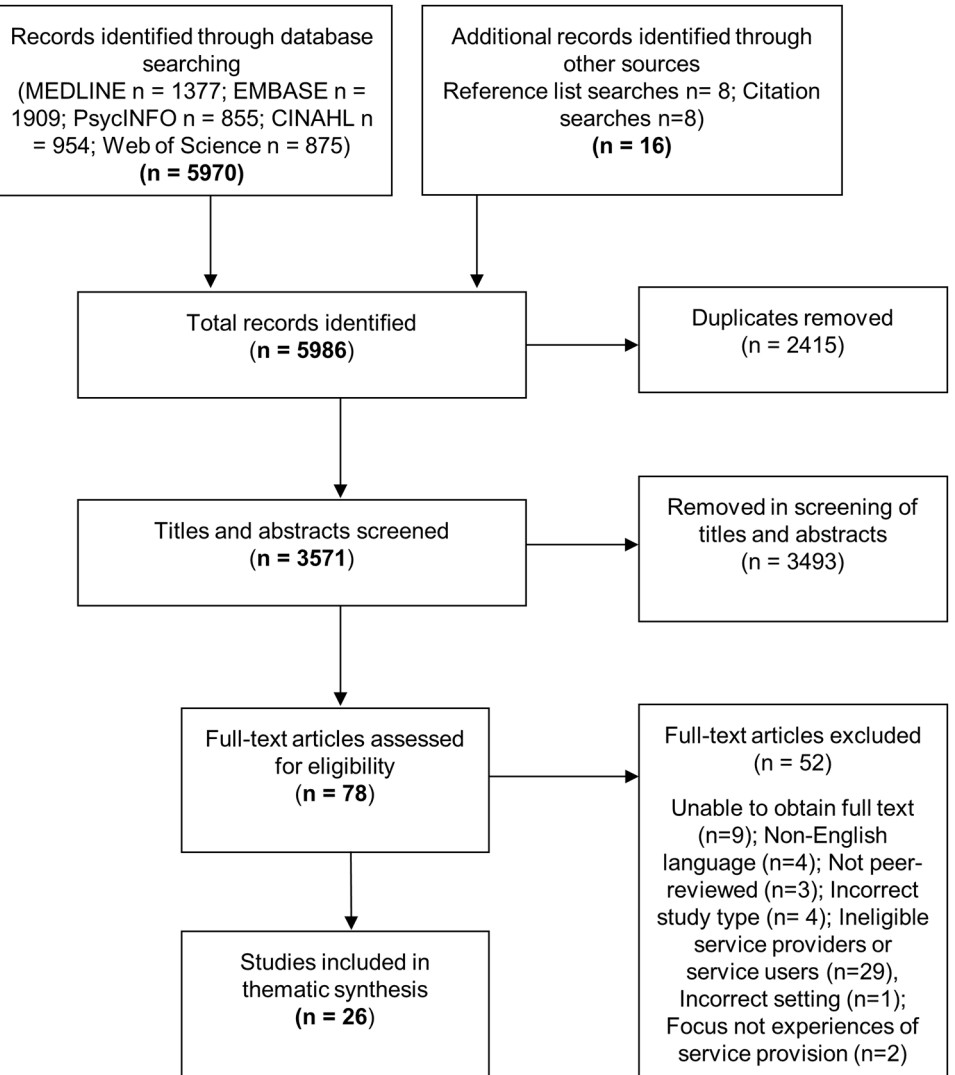

**Figure 1** Flow diagram of systematic search and study selection.

## Characteristics of included studies

The 26 articles included were based on 21 primary studies, of which 19 were qualitative studies[17 18 37–58] and 2 were mixed methods.[19 59] Nine articles were from Australia,[41 43 45–48 50 57 58] seven from the UK,[18 37–40 42 59] three from the Netherlands[44 52 53] and one from each of Denmark,[17] Switzerland,[19] New Zealand,[49] Sweden,[51] the USA,[54] Ireland[55] and Canada.[56] All articles were published between 1999 and 2016. Service users were described as 'refugees' in 11 articles,[17 41 43–49 54 56] 'asylum seekers' in 6 articles,[19 38 52 53 55 59] 'refugees and asylum seekers' in 5 articles,[18 37 39 40 42] 'of refugee background' in 3 articles[50 57 58] and 'involuntary migrants' in 1 article.[51]

Qualitative data extracted for this synthesis were derived from 357 participants with a combined sample of 194 nurses, 35 midwives and 128 doctors. None included pharmacists. Data collection methods varied across the 21 primary studies represented, with 14 solely using individual interviews (including in-depth, semi-structured, unstructured),[17–19 37–40 42 44 45 47–49 51 55 56 59] one employed group interviews only[54] and four combined individual

and group interviews.[43 46 50 57 58] One study used observational methods and individual interviews,[41] and one combined group interviews and qualitative questionnaires.[52 53] Table 1 summarises characteristics of included articles and online supplementary file 3 contains the complete data extraction.

## Quality assessment

Application of the CASP critical appraisal tool revealed variable results across the 26 articles assessed. All except one article[39] gave a clear statement of the research aims. The majority (21 articles)[17–19 37–40 42–44 46 48 49 51–58] sufficiently described the sampling strategy and provided some rationale for participants' selection. Possible reasons for non-participation were discussed in only four articles.[19 37 46 47] The data collection method was stated in all articles, however, the extent of information provided about interview schedule's content was variable. A significant number did not describe the setting of data collection (13 articles)[38 43–47 50 52 53 56–59] or the identities of interviewers (12 articles).[18 19 38–42 46 47 55 58 59] Only eight

**Table 1** Characteristics of articles included in the thematic synthesis

| First author | Publication year | Country | Eligible participants* and practice setting | Service users† | Data collection method | Analysis methodology | Study aims/objectives‡ |
|---|---|---|---|---|---|---|---|
| Begg and Gill[37] | 2005 | UK | Seventeen general practitioners (GPs) General practice | Refugees and asylum seekers | Semi-structured interviews | Thematic framework | To identify some of the concerns of 17 GPs working in an urban environment |
| Bennett and Scammell[38] | 2014 | UK | Ten midwives Community, rotational, specialist and delivery suite midwives | Female asylum seekers | Semi-structured interviews | Thematic analysis | To gain an in-depth analysis of the experiences of midwives and their understanding of the specific needs of asylum-seeking women. The findings would be used to inform education, practice and policy to enable more effective delivery of woman-centred care for this group locally |
| Burchill[39]§ | 2011 | UK | Fourteen health visitors London borough | Refugees and asylum seekers | In-depth interviews | Framework | Not clearly stated |
| Burchill and Pevalin[18]§ | 2012 | UK | Fourteen health visitors London borough | Refugees and asylum seekers | In-depth interviews | Framework | To determine the barriers to effective practice that health visitors when working with refugees and asylum seekers |
| Burchill and Pevalin[40]§ | 2014 | UK | Fourteen health visitors London borough | Refugees and asylum seekers | In-depth interviews | Framework | Explored the experiences of health visitors working with refugee and asylum-seeking families in central London, and assessed the dimensions of their cultural competency using Quickfall's model |
| Carolan and Cassar[41] | 2008 | Australia | Two midwives African women's clinic (community health centre) | Female African refugees | Observational methods and semi-structured interviews | Thematic analysis | To explore factors that facilitate or impede the uptake of antenatal care among African refugee women |
| Crowley[59]¶ | 2005 | UK | Ten GPs General practice | Asylum seekers | Telephone interviews | Not specified | To assess the mental healthcare needs of adult asylum seekers in Newcastle upon Tyne |
| Drennan and Joseph[42] | 2005 | UK | Thirteen health visitors Two London boroughs | Refugees and asylum seekers | Semi-structured interviews | Framework | Describe health visitors' experiences working in Inner London and identifying and addressing the health needs of refugee woman in the first 3 months after the birth of a baby. Investigate health visitors' perceptions of effective and ineffective strategies in identifying and addressing health needs of these women. Investigate whether health visitors used a framework corresponding to Maslow's theory of a hierarchy of needs to prioritise their public health work |

Continued

**Table 1** Continued

| First author | Publication year | Country | Eligible participants* and practice setting | Service users† | Data collection method | Analysis methodology | Study aims/objectives‡ |
|---|---|---|---|---|---|---|---|
| Farley et al[43] | 2014 | Australia | Twenty GPs Five practice nurses General practice | Newly arrived refugees | Focus groups and semi-structured interviews | Thematic analysis | Explored the experiences of primary healthcare providers working with newly arrived refugees in Brisbane…focusing on the barriers and enablers they continue to experience in providing care to refugees |
| Feldmann et al[44] | 2007 | The Netherlands | Twenty-four GPs General practice | Refugees (Afghan/Somali) | In-depth interviews | Thematic analysis | To confront the views of refugee patients and GPs in the Netherlands, focusing on medically unexplained physical symptoms |
| Furler et al[45**] | 2010 | Australia | Eight family physicians Community health centre | Refugees with depression | Semi-structured interviews | Thematic analysis | This study explores the complexities of this work (clinical care for depression) through a study of how family physicians experience working with different ethnic minority communities in recognising, understanding and caring for patients with depression |
| Griffiths et al[46] | 2003 | Australia | Thirteen nurses Two nurse managers Refugee reception centre | Refugees | Focus groups and semi-structured interviews | Thematic analysis | To identify the skills, knowledge and support nurses require to provide holistic and competent care to refugee children and their families and the nature of support that is required to assist their transition back to mainstream health services |
| Jensen et al[17] | 2013 | Denmark | Nine GPs Medical clinics | Refugees | Semi-structured interviews | Content analysis | To qualitatively explore issues identified by GPs as important in their experiences of providing care for refugees with mental health problems |
| Johnson et al[47] | 2008 | Australia | Twelve GPs General practice | Refugees | Semi-structured interviews | Template analysis | To document the existence and nature of challenges for GPs who do this work in South Australia. To explore the ways in which these challenges could be reduced. To discuss the policy implications of this in relation to optimising the initial healthcare for refugees |
| Kokanovic et al[48**] | 2010 | Australia | Five GPs Community health centre | Refugees with depression | In-depth interviews | Thematic analysis | We explore a set of cultural boundaries across which depression is contested: between recent migrants to Australia from East Timor and Vietnam, and their white 'Anglo' family doctors |

Continued

**Table 1** Continued

| First author | Publication year | Country | Eligible participants* and practice setting | Service users† | Data collection method | Analysis methodology | Study aims/objectives‡ |
|---|---|---|---|---|---|---|---|
| Kurth et al[19]¶ | 2010 | Switzerland | Three physicians Three nurses/midwives Women's clinic | Female asylum seekers | Semi-structured interviews | Grounded theory | To investigate the reproductive healthcare provided for women asylum-seekers attending the Women's Clinic of the University Hospital in the city of Basel, Switzerland. To identify the health needs of asylum seekers attending the Women's Clinic and to investigate the healthcare they received in a health maintenance organisation) specifically established for asylum seekers. Explored the perceptions of the healthcare professionals involved, about providing healthcare for this group in this setting |
| Lawrence and Kearns[49] | 2005 | New Zealand | Five medical practitioners Community health centre | Refugees | In-depth interviews | Thematic analysis | This paper reports on research that sought to reveal the barriers faced by refugees in accessing health services, and the challenges faced by providers in endeavouring to meet needs in an effective and culturally appropriate manner |
| Riggs et al[50] | 2012 | Australia | Twelve nurses Maternal and child health services | Refugee background mothers | Focus groups and Interviews | Thematic analysis | To explore the utilisation and experience of maternal and child health services in Melbourne, Victoria for parents of refugee background from the perspective of users and providers |
| Samarasinghe et al[51] | 2010 | Sweden | Thirty-four primary healthcare nurses Various: maternity, child, school, community healthcare, nurse-led clinics | Involuntary migrant families | Interviews | Contextual analysis | The aim of this study was to describe the promotion of health in involuntary migrant families in cultural transition as conceptualised by Swedish primary health care nurses |
| Suurmond et al[53]†† | 2013 | The Netherlands | Thirty-six nurse practitioners Ten public health physicians Asylum seeker centres | Newly arrived asylum seekers | Group interviews | Framework | To describe the tacit knowledge of Dutch healthcare providers about the care to newly arrived asylum seekers and to give insight into the specific issues that healthcare providers need to address in the first contacts with newly arrived asylum seekers |
| Suurmond et al[52]†† | 2010 | The Netherlands | Eighty-nine nurse practitioners (questionnaires) Thirty-six nurse practitioners (group interviews) Asylum seeker centres | Asylum seekers | Questionnaires and group interviews | Framework | We explored the cultural competences that nurse practitioners working with asylum seekers thought were important |

Continued

**Table 1** Continued

| First author | Publication year | Country | Eligible participants* and practice setting | Service users† | Data collection method | Analysis methodology | Study aims/objectives‡ |
|---|---|---|---|---|---|---|---|
| Tellep et al[54] | 2001 | USA | Six school nurses / Schools | Refugees | Focus group | Unspecified | To describe the nature and meaning of school nurses' and Cambodian liaisons' experiences of caring for Cambodian refugee children and families and to explore whether those meanings validated Dobson's conceptual framework of transcultural health visiting |
| Tobin and Murphy-Lawless[55] | 2014 | Ireland | Ten midwives / Maternity hospitals | Female asylum seekers | In-depth unstructured interviews | Content analysis | To explore midwives' perceptions and experiences of providing care to women in the asylum process and to gain insight into how midwives can be equipped and supported to provide more effective care to this group in the future |
| Twohig et al[56] | 1999 | Canada | Six family practice nurses / Ten family physicians / Clinic at refugee processing centre | Refugees | Semi-structured interviews | Textual analysis | To explore roles of family physicians and family practice nurses who provided care to Kosovar refugees at Greenwood, Nova Scotia |
| Yelland et al[57]§ | 2014 | Australia | Ten midwives / Maternity services | Refugee background families | Interviews and focus groups | Thematic analysis | (1) investigate Afghan women and men's experience of the way that health professionals approach inquiry about social factors affecting families having a baby in a new country and (2) investigate how health professionals identify and respond to the settlement experience and social context of families of refugee background |
| Yelland et al[58]§ | 2016 | Australia | Ten midwives / Maternity services | Refugee background families | Interviews and focus groups | Thematic analysis | (1) Describe Afghan women's and men's experiences of language support during pregnancy check-ups, labour and birth; (2) explore health professionals' experiences of communicating with Afghan and other refugee clients with low English proficiency and (3) consider implications for health services and health policy |

*Some studies included some participants not eligible for this review. These participants have not been included in this table.
†Service users as described by the authors.
‡The aims and objectives are from the author (ie, extracted directly from papers).
§These articles were based on data from the same sample, but reported different aspects.
¶Mixed methods were used in these studies. This table only includes characteristics of the qualitative element relevant to this review.
**The five GPs in Kokanovic 2010 are included within the eight physicians in Furler et al,[45] but report different aspects.
††The 36 nurse practitioners are common between articles, but report different aspects.

articles[43 47–50 52 56 59] gave justification for chosen data collection methods or interview settings. Data saturation was rarely discussed, featuring in five articles.[37 43 47 48 56]

Reflexivity was particularly poorly discussed across articles. Only seven[37 39 43 48 51 54 55] reflected on potential bias and influence of researchers at any stage in the study (formulation of review question, sampling, data collection or analysis).

Ethical approval was described in the majority of articles (23 articles),[17–19 37–43 45–48 50–58] but they often lacked sufficient information to judge whether ethical standards had been followed. Thirteen articles[17–19 38–43 48 51 53 55] described how participants were informed about the nature and purpose of the study, 17 articles[17 19 37 38 40 42 43 48–56 60] described obtaining consent and 12 articles[17 37 41–43 46 47 51–55] discussed how confidentiality was assured or maintained.

The approach to data analysis was described to some extent in all but one article[59]; however, there was variation in the level of detail given. Involvement of multiple researchers in the analysis process was reported in 19 articles.[17–19 37 39–43 45–48 50 51 55–58] The majority (25 articles)[17–19 37–58] gave support for findings with references to primary data (eg, quotations from participants). Findings were generally clearly presented and discussed in context of wider research literature, policy and practice, although a few (six articles)[39 40 42 49 54 56] were limited in this area. Ten articles[19 37 41 43 45 50–53 57] explicitly reflected on the credibility of their findings.

Full details of the CASP assessment are provided in online supplementary file 4.

## Thematic synthesis findings

Challenges and facilitators for health professionals providing primary healthcare to refugees and asylum seekers were interpreted within 11 descriptive themes, embedded in 3 analytical constructs: healthcare encounter (trusting relationship, communication, cultural understanding, health and social conditions, time), healthcare system (training and guidance, professional support, connecting with other services, organisation, resourcing and capacity) and asylum and resettlement. Figure 2 illustrates the relationships between analytical constructs and descriptive themes. Healthcare encounters occur within the environment of healthcare systems, both of which operate within wider asylum and resettlement policies and processes. Table 2 provides a taxonomy of challenges and facilitators and table 3 contains illustrative quotations from primary studies for each descriptive theme.

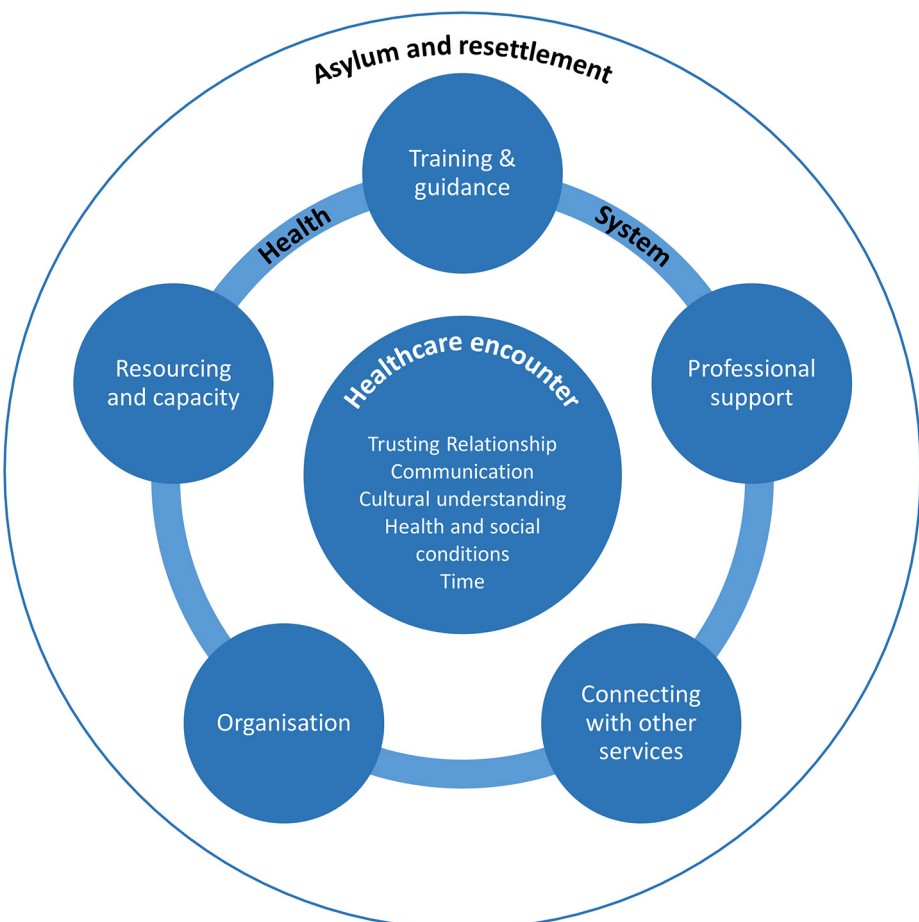

**Figure 2**  Model illustrating analytical constructs and descriptive themes.

**Table 2** Taxonomy of challenges and facilitators

| Analytical construct | Descriptive theme | Challenges | Facilitators |
|---|---|---|---|
| Healthcare encounter | Trusting relationship | -Transience of refugees/ asylum seekers<br>-Suspicion of authorities | -Continuity of care<br>-Assisting with wider needs<br>-Taking an interest<br>-Compassion/empathy<br>-Explaining role |
| | Communication | -Language: assessing case history/gaining consent/ensuring patient understanding<br>-Interpreters: additional time/expense, unavailability, inaccuracy/imposition of own views<br>-Telephone interpreters: impersonal, technological failures<br>-Illiteracy<br>-Lack of language specific resources | -Interpreters: professionally trained, continuity<br>-Telephone interpreters: increased availability<br>-Visual aides |
| | Cultural understanding | -Different understandings of health concepts/terminology/healthcare systems<br>-Understanding patient's symptoms<br>-High expectations of patients<br>-Different cultural values | -Knowledge of other cultures: values, health practices, body language<br>-Personal qualities: sensitivity, empathy, cultural humility |
| | Health and social conditions | -Physical: communicable diseases, female genital mutilation, injuries<br>-Unusual diseases<br>-Psychological: torture, abuse, social difficulties, somatisation<br>-Lacking skills, knowledge, support | -Training<br>-Guidance<br>-Professional support<br>-History taking |
| | Time | -Increased time requirement<br>-Increased duration/occurrences of appointments<br>-Insufficient time: rushed appointments<br>-Time taken away from other patient groups | |
| The healthcare system | Training and guidance | -Lack of training/guidance<br>-Lack of awareness of available resources<br>-Time constraints | -Cultural competency training<br>-Orientation to services/resources/asylum process<br>-Culture-specific information |
| | Professional support | -Deficiency of professional support<br>-Supporting traumatised patients without support<br>-Isolation | |
| | Connecting with other services | -Referral difficulties; services not present/not suitable<br>-Difficulty understanding/navigating healthcare system | -Establishing referral pathways: health system/civil society<br>-Accompanying refugees and asylum seekers<br>-Communication/coordination/collaboration with other services<br>-Codelivery of services<br>-Multiagency teams |
| | Organisation | | -Flexibility of primary healthcare system: innovation/adaptation<br>-Specialised services |
| | Resourcing and capacity | -Increased costs<br>-Funding shortages<br>-Workforce shortages<br>-Inflexibility/unsuitability of interpreter services | |
| Asylum and resettlement | Asylum and resettlement | -Policy restrictions<br>-Conflicts of interest<br>-Understanding changing policy environment and healthcare provisions<br>-Perceived abuses of system | -Training in asylum and resettlement policy/process |

**Table 3** Illustrative quotations

| Theme | | Quotation and reference* |
|---|---|---|
| **Healthcare encounter** | | |
| Trusting relationship | Challenge | "… you put your mind around trying to sort things out, the dreadful things that have happened to them, and then the next week it will be a different family there and you start the whole process all over again, trying to build up some sort of trust… "[42] |
| | Facilitator | "Creating trust is an important aspect, to show that you are interested in the person, not only in the disease; to show that you want to know something about the context. Sometimes it is difficult to find time for it in a busy practice, but I see it is a worthwhile investment".[44] |
| Communication | Challenge | "I've had some pretty bad examples recently of interpreters where they have actually started giving their opinion, which has been a nightmare, …they start adding their points of view".[40] |
| | | "The phone interpreter is too impersonal. And I found that a lot of them use mobile phones so you're constantly cutting out… "[50] |
| | Facilitator | "Everything comes down to communication. To know what's going on, what they need, what you need, because it's a partnership, isn't it?"[38] |
| | | '… this (telephone interpreting) is available 24 hours and is instantaneous… it is revolutionised, all the doctors use it, the receptionists, the nurses… '[37] |
| Cultural understanding | Challenge | '…they have a different culture, so their cultural perception of symptoms and what they mean… trying to interpret the difference between a bloated abdomen and a painful abdomen, just becomes an impossible task'.[43] |
| | | "I sometimes say, 'I am only a doctor'. Sometimes there are far greater expectations than what you can honour"[17] |
| | | "I think most people understand sadness…but in some cultures, they don't understand (depression) as a condition that requires treatment"[45] |
| | Facilitator | '…there were specialised nurses who had worked overseas, who gave workshops for us, and explained much of the history, and explained some of the conflicts which they bring over here'.[54] |
| Health and social conditions | Challenge | "I am quite overwhelmed at times as to how complex these ladies' lives are… "[38] |
| | | "I guess it is out of our comfort zone, because our medical experience doesn't include the exotic illnesses that they front up with… "[47] |
| | | "Midwives spoke of the emotional impact of working with women with trauma histories: 'How does it affect me, you just feel sad you know, but you just do the best that you can and that's all you can do "[55] |
| | Facilitator | "(Specialist team teaching sessions) is the sort of thing that people need to help give them a baseline of knowledge, and I suppose, the support to realise that there are other people they can talk to, to help them and signpost, or help them to signpost their clients in the right direction".[40] |
| | | 'We do not need to know the whole lot; we do not need the whole case history […] to have a bit more understanding'.[38] |
| Time | Challenge | '… generally speaking a consultation with a refugee will take twice as long (as with) a local patient'.[37] |
| | | '…providing care with interpreters was more time consuming than without, meaning that midwives had to "juggle their time" to facilitate good care'.[38] |
| **Healthcare system** | | |
| Organisation | Facilitator | 'The flexibility of the general practice setting enabled providers to act on their commitment to provide refugee healthcare, allowing them to be responsive and innovative in their approach to caring for refugees and also providing flexibility in the hours they work'.[43] |
| | | 'Participants felt that significant gains had been made to the refugee healthcare system, with the establishment of a specialised service. One provider working in the field for some time described thinking… fantastic, finally'[43] |
| Resourcing and capacity | Challenge | '… (asylum seekers) should be budgeted for… they are actually slightly harder work than somebody else (this) needs to be acknowledged'.[37] |
| | | "But I was more angry that I just needed more hands to help. So, for me it was about practical support".[46] |

Continued

| Theme | | Quotation and reference* |
|---|---|---|
| Training and guidance | Challenge | "*Even when we called… the* [Division of General Practice]… *they didn't know how to guide us… I think we didn't have a guideline…* "[43] |
| | Facilitator | 'The specialist team facilitated a rolling programme of training for frontline staff working with refugees and asylum seekers, and this was regarded as an effective way of sharing knowledge'.[39] |
| Professional support | Challenge | '…lack of institutional support all contributed to varying feelings of powerlessness on the part of the midwives themselves'.[55] |
| | Facilitator | 'They described the value of currently available external supports, including language classes, translation and interpreting services and specialised refugee health services, particularly in the area of mental health'.[43] |
| Connecting with other services | Challenge | "*She explained she had seen a lot of problems…I put her touch with a voluntary (nationality specific) counselling organization to then discover she had to pay and she can't afford it*".[42] |
| | Facilitator | "*So I referred her to… and we went together for a joint meeting… FORWARD [a women's campaign and support charity] specialises in FGM and I set her up for an appointment there and she was referred to a specialist nurse… who was able to look at potentially reversing part of the FGM and the client was happy for this to happen and actually did attend*".[40] |
| **Asylum and resettlement** | | |
| | Challenge | '*These requirements differed: on the one hand to be the caregiver, to be the patient's advocate in fact, and on the other to act as advocate of the Federal Office for Refugees, and thirdly to be responsible for the organisation, to save costs for the health insurance. But that is simply not possible*'.[19] |
| | | "*I don't know if there is some sort of system that they go through, or some sort of protocol that they, medically, have to go through before they are granted visas…*"[47] |

*Participant's quotations are in italics, study authors text is normal typeface.

## The healthcare encounter

Challenges and facilitators for healthcare provision to refugees and asylum seekers were experienced within the healthcare encounter. This is the milieu of personal engagement between health professionals and service users. Five inter-related factors influenced health professionals' practice: trusting relationship, communication, cultural understanding, health and social conditions and time.

### Trusting relationship

Building trusting relationships with refugees or asylum seekers featured in 15 of the articles.[18 40–42 44–46 48 50–55 57] Facilitators included continuity of the attending care provider[42 50 52 54 57]; taking an active interest in their background, language and culture[40 44 54 57] and assisting them with their wider needs.[18 40 50] Having a compassionate and empathetic disposition was also seen as important in relationship building.[41 51 52 55 57] The transient nature of some service users made building relationships challenging[42] and trust was threatened when refugees or asylum seekers thought that healthcare professionals were associated with immigration authorities.[38 42 52] Health professionals found that clearly explaining their role and confidentiality brought reassurance and allayed suspicions.[42 52] Some benefits of establishing trusting relationships were said to be increased engagement with the healthcare service by refugees and asylum seekers,[18 40 41 50] and greater levels of disclosure about their health and social concerns.[42 48 50 52 57]

### Communication

Communication was a theme found in 22 included articles.[17–19 37 38 40 42–45 47–58] The language barrier was widely cited as challenging while caring for refugees and asylum seekers.[17–19 37 38 40 43 44 47–51 53–58] Individual articles elaborated that language barriers presented difficulties in assessing case histories,[19] gaining consent[55] and ensuring patients understood treatment.[42]

Using interpreters was considered a major facilitator in communication[17 37 38 40 43 45 50 51 57] and was maximised when interpreters were well-trained and familiar with medical terminology.[17 45] Continuity of the interpreter was deemed important in fostering good communication and increased confidence in the integrity of translation.[38 40 45 50] There were, however, challenges associated with interpreter use.[17–19 37 38 40 42 43 45 47 49 50 52 55 56 58] Communicating through interpreters required additional time[38 47] and financial expense.[55] Suitable interpreters were not always available at the appropriate time,[17 38 42 43 47 55] which could lead to delayed, extended or rearranged appointments.[17 38 47] This led, in some cases, to family or other community members being asked to translate instead of professional interpreters.[42 55] Participants were also concerned that interpreters did not always accurately communicate[37 40 43 45 55 56] and may impose their own views.[40 43] The use of telephone interpreters received mixed opinions. Advocates welcomed the increased availability of interpreters at any time of the day,[37] but others

felt they were more impersonal[50 58] and pointed to technological failures that hindered communication.[50 58]

Further communication challenges included unavailability of written health information in service users' languages[53 57] and in some cases patients were unable to read or write.[43] To improve communication with those with limited language skills, some participants used objects or other visual aids.[51]

### Cultural understanding

Cultural understanding was a theme described across 21 articles.[17 18 37 38 40–49 51–57] Healthcare provision could be challenging, when there were different understandings of health, illness or healthcare.[17 18 40 44–49 51 53 55] Health literacy could be limited[43 47 53] and different terms could be used to refer to health conditions.[18 45 48 57] Healthcare concepts such as preventative care (eg, screening),[47 49] mental healthcare[48 57] and self-management[51] were sometimes unfamiliar. Service users also lacked understanding of host countries' healthcare systems,[37 40 42 43 45 49] making them prone to miss appointments,[43] and attempt to inappropriately access services.[37]

Differences in health culture presented difficulties for health professionals' understanding of patient's symptoms[45] and required additional time and effort explaining health conditions, healthcare concepts or health systems.[42 47 51] It was also reported that some refugees or asylum seekers had very high, and sometimes unrealistic, expectations of health services or health professionals,[17 37 40 52 53] which needed to be counteracted by participants.[17 53] Disparities in cultural values such as gender roles, decision-making, social taboos and time-orientation were also mentioned as challenges,[41 47 48 53] with some health professionals expressing uncertainty about approaching some clinical tasks such as physical examinations.[47]

Gaining knowledge and understanding about cultures of refugees and asylum seekers was viewed as an important facilitator in cross-cultural care.[38 40 42 47 52 54 55 57] This included understanding differences in values,[42] body language,[52] health practices[42] and health presentations.[52] Cultural understanding allowed health professionals to adjust their healthcare practice accordingly.[40 45 48 49 51 55 56] Personal qualities in health professionals that were deemed to enhance cross-cultural interactions were sensitivity,[49 52 54] empathy[40 41 54] and cultural humility.[54 55]

### Health and social conditions

Health professionals spoke of challenges in dealing with physical, psychological and social problems that were typically presented by refugees and asylum seekers.[17 37 40 43 44 46 47 52 53 55–57]

Physical conditions presented challenges[37 40 43 44 47] and included tropical diseases such as malaria and schistosomiasis[43]; other communicable diseases such as TB and HIV[37 40 44]; and nutritional deficiencies.[37 40 44] Physical injuries were also encountered, such as female genital mutilation (FGM)[40 55] and injuries inflicted from conflict or torture.[40] Health professionals did not

always feel prepared or equipped to deal with these conditions[43 47] and there were concerns from general practitioners that some conditions could remain undiagnosed.[43 44 47]

Psychological conditions were considered challenging to deal with,[17 37 40 43 46 52 53 55–57] and were frequently seen among refugees and asylum seekers.[37 43] These included psychological trauma related to war,[17] torture[40 43 46] and other abuses.[17 38 40] Postmigration stresses were also perceived to impact negatively on their mental health such as the asylum and resettlement process,[17 40 47] social isolation[17 45 55] and other social vulnerabilities.[40 50 57] Health professionals found engaging with these service users emotionally difficult[37 55] and distressing when hearing their disturbing stories.[40 42 46 55] They also expressed feelings of powerlessness[17 46 55] believing they lacked required skills, knowledge and support to respond to their complex psychological needs.[43 57]

A further challenge noted by health professionals across four articles was the manifestation of medically unexplained symptoms (somatisation) among some refugees and asylum seekers,[18 43 44 48] which could be frustrating[43] and time consuming to address.[43 48]

Several facilitators were identified that could help deal with complex physical and psychological conditions. Careful history-taking of medical, social and migration background was helpful[38 44 50 53 57] and could identify possible risk factors.[53 57] Training in conditions common among refugees and asylum seekers was deemed valuable,[37 38 40 46 52 53 55] increasing confidence in care delivery[40] and resulting in 'more effective, evidence-based care'.[38] Clinical guidelines for refugee healthcare were considered beneficial,[37 47] although these were often unavailable.[37 47] Professional support was regarded as a facilitator,[37 38 42 43 46 51 55] provided within services[42] or from external organisations specialising in refugee healthcare.[43 46] The importance of psychological support for those working with traumatised patients was highlighted,[46 51 55] such as counselling or debriefing.[46 51] Challenges around training, guidance and professional support are described in 'The healthcare system' section.

### Time

A significant challenge faced by health professionals was the time required to provide healthcare for refugees and asylum seekers.[18 37 38 40 43 47 49–51 55 56 59] More time was necessary due to the aforementioned challenges around building relationships,[18 38 40] communication,[38 50 55 59] achieving cultural understanding[47] and dealing with complex health conditions.[18 38 47 50 51] This additional time demand meant that appointments needed to be extended in duration[37 47] or occur more frequently.[18 49] Health professionals were concerned that time limitations could lead to 'rushed consultations'[59] and the potential to miss some conditions.[59] Some also commented that the extra time spent caring for refugees and asylum seekers drew them away from other patient groups.[40 43]

## The healthcare system

Health systems have been defined as 'the combination of resources, organisation, financing and management that culminate in the delivery of health services to the population'.[61] They are the environment in which healthcare encounters take place. Healthcare professionals described health system-related challenges and facilitators within five areas: training and guidance, professional support, connecting with other services, organisation and resourcing and capacity.

### Training and guidance

As already described in 'health and social conditions', health professionals thought that specific training and guidance would facilitate their clinical practice, improving their competence and confidence. Positive examples of training delivered were: orientation to services and resources available for refugees and asylum seekers[40]; culture-specific information[42 54]; engaging with women about FGM[40] and trauma-sensitive care.[46] Despite this, a broad base of participants identified lack of training, education or guidance as detrimental to practice.[17 37 38 42 43 46 49 50 55] Even when available, training may be inaccessible due to lack of awareness or time constraints.[43] Participants called for more training, guidance or information regarding integration with other health and social care services,[37 42 50] health profiles of specific groups,[46] cultural awareness/competence[42 46 47 49 50] and the wider process of asylum.[37 42]

### Professional support

As reported in the earlier section 'Health and social conditions', professional support was needed by health professionals working with refugees and asylum seekers. However, professional support was identified as deficient in healthcare systems.[37 43 46 55] Participants in one study described 'isolation'[43] that they felt within the healthcare system and another study described support networks as 'non-existent'.[37] Concerns were raised that health professionals exposed to distressing stories were not provided with sufficient psychological support.[46 55]

### Connecting with other services

Connecting with other health and social care services was another important facilitator for health professionals.[17 18 38 40 42 47 49–52 54] Establishing referral pathways to different services in the healthcare system[40 42 47 51 52] and services within civil society[40 42 47] could direct refugees and asylum seekers to appropriate care. Some health visitors described accompanying refugees and asylum seekers to support groups to help with introductions.[40 42] Good communication and cooperation between services was helpful[38] and fruitful collaborations with other services were recognised, such as delivering services together[50 51] and working in multiagency teams to deliver holistic healthcare.[38 51 54]

Health professionals spoke of some difficulties referring refugees and asylum seekers to other health or social services.[17 18 39 40 50 55] Some, services were not set up to meet their needs,[17 40] others would not receive referrals because they were operating at full capacity[18 39] and sometimes services were simply not present.[18 55] These challenges could be accentuated when health professionals found it difficult to navigate complex healthcare systems themselves.[43]

### Organisation

Some articles highlighted flexibility in primary healthcare systems as beneficial for practice among refugees and asylum seekers.[40 41 43 49 50] This allowed for innovative approaches to optimise service delivery[40 43] such as relocating services to more accessible places[18 40 42 50] and adaptation of working patterns to better suit service users' needs.[43 50]

Provision of specialised services for refugees and asylum seekers was supported across some studies,[37 40 43 47] including initial health assessment services,[47] specialist teams[40 47] and specialist centres.[37 47] However, it was emphasised that these should integrate well into mainstream healthcare services.[37 40]

### Resourcing and capacity

Longer, more frequent appointments and utilisation of interpreters led to additional costs being incurred,[18 19 37 43 47 49 51] which some felt was not taken into account in health system financing models.[43 47 49] Some participants did not think that they could deliver adequate care as a result of funding shortages,[37 55] with one study citing an example where interpreters were not able to be used because of lack of finance.[55]

Shortages in workforces were reported in some articles,[46 47 49] putting additional workload and stress onto health professionals.[46 49] Reported consequences of this were closures of services to new patients[47 49] and health professionals leaving their posts, further exacerbating the problem.[49] Interpreter shortages were also mentioned as a difficulty[46 49 56] along with inflexibility of their service operations.[37 42 55]

### Asylum and resettlement

Further challenges were associated with the immigration status of, and legislative policy towards, refugees and asylum seekers.[18 19 37 39 40 46 47 59] In some instances, health professionals were hindered in meeting health needs due to policy restrictions.[40] Difficulties understanding the frequently changing policies towards, and entitlements for, refugees and asylum seekers were reported[39 40] and uncertainty was expressed about healthcare pathways for this group on arrival in the host country.[47] Some health professionals described conflicts in their professional duty to act as an advocate for their patients while requirements were placed on them to conduct assessments used to inform the asylum process.[19 46] Another concern raised was a perception that service users were abusing the health and welfare systems,[18 37 40 59] such as feigning symptoms of post-traumatic stress disorder to further their asylum claims[37] or illegal benefit claims.[18]

## DISCUSSION

Three analytical constructs containing 11 descriptive themes were interpreted in the thematic synthesis. Challenges and facilitators were located within the healthcare encounter (trusting relationships; communication; cultural understanding; health and social conditions; time), working within the healthcare system (training and guidance; professional support; connecting with other services; organisation; resourcing and capacity) and asylum and resettlement.

The growing research field of 'cultural competence' identifies components that can be incorporated into practice to enhance quality of care towards ethnic minority groups and reduce healthcare inequalities.[62 63] Betancourt et al[62] defined cultural competence in healthcare as 'the ability of systems to provide care to patients with diverse values, beliefs and behaviours, including tailoring delivery to meet patients' social, cultural and linguistic needs'.[62] This literature mirrors themes interpreted in the current review, including trusting relationships, communication and cultural understanding, as key components that may be optimised to improve healthcare and reduce inequalities.[62 63]

Trusting relationships are essential for effective healthcare delivery.[64–66] Murray et al[67] identified continuity of relationship, time, interpersonal skills and 'getting to know patients' as enhancers of trust between health professionals and patients. The current review likewise recognised these elements, and it can be argued that even greater attention to trust-building is needed for refugees and asylum seekers, a vulnerable and ethnically diverse group who may be apprehensive about engagement with healthcare systems.[68 69]

Communication between health professionals and patients is also regarded as essential.[70] Language discordance may compromise the quality of healthcare, lessening detection of ill health and referral to further healthcare.[71 72] Health professionals in the current review consistently thought language barriers hindered their work with refugees and asylum seekers. The main strategy used to overcome language barriers was communication through interpreters, as is recommended in the wider literature.[73–75] However, concerns were raised about the quality and availability of interpreters. Generally, it is recommended that professional interpreters are used, as they have been trained in professional standards, medical terminology and ethical issues.[75] Ad hoc interpreters such as family or community members may be used pragmatically, although this may diminish the quality of interpretation and threaten patient confidentiality.[74 75] Remote interpretation, such as telephone or video services have been developed to provide more efficient and timely services.[76 77] The merits of such services have been debated[76 77] and conflicting opinions were likewise given in this review. A systematic review[77] reported no significant difference in patient and provider satisfaction between remote and face-to-face interpreters, although

subsequent primary studies have suggested a significant preference for in-person interpreters.[76]

Consistent with other research,[6–8 11–13] health professionals encountered challenges dealing with complex physical, psychological and social problems of refugees and asylum seekers and did not always feel prepared to meet their needs. They also reported challenges in cross-cultural care such as different understandings of health, healthcare and healthcare systems, which introduced complications.

Participants in this review saw opportunities for improving care by working together with other health services and civil society. Identifying these organisations and possible areas of collaboration such as information sharing, referral pathways and joint service delivery may benefit health providers, health professionals and service users.

The organisation and delivery of primary healthcare services to refugees and asylum seekers is a growing research area, with service models being developed that integrate specialised components with existing structures.[78 79] A model innovated in Australia established 'Beacon practices', which have expanded capacity for refugee care and may flexibly resource local services.[79] Such integrated services provide specialised resources without isolating refugees and asylum seekers from general practice, which was a concern raised by some participants in this review.

Health professionals and health services operate within, and are influenced by, the wider healthcare policy environment. Decisions made at a political and health system levels invariably impact on frontline clinical practice in areas such as resourcing priorities, health professional roles and healthcare access.[80] Health professionals in this review recognised associated challenges, particularly when healthcare pathways were unclear and changeable. This emphasises the need for policy makers to provide consistent, clear and up-to-date guidance on asylum and resettlement health policy for health professionals.

### Public health implications

A central concern in public health is reduction of inequalities in health and healthcare.[81 82] WHO has established a commission on the social determinants of health to support countries and recommend actions that address inequalities in health.[82] Healthcare inequalities exist when certain groups systematically receive lower quality care than the general population, resulting in poorer health outcomes.[80 83] These inequalities have been widely observed in healthcare provision to ethnic minority groups across a broad range of health services[80] and has been highlighted as an issue for refugees and asylum seekers in the UK.[21] However, through knowledge translation, where evidence is moved into practice, challenges and facilitators identified in this review may be mapped onto components of healthcare interventions that may minimise such healthcare inequalities.[84]

Reduction in healthcare inequalities will likely require targeting healthcare resources towards disadvantaged groups.[79] For example, health professionals in this review highlighted the need for additional resources such as interpreter services, training and professional support to improve quality of care for refugees and asylum seekers.

## Recommendations

### Practice

Health professionals should be sufficiently resourced to meet the complex needs of refugees and asylum seekers. This should include provision of appropriate training on areas of cultural competence, asylum policies and process and health conditions. It is recommended that specific clinical guidelines are developed for provision of care to refugees and asylum seekers, drawing on the best available evidence. Further professional support should be given to those working with patients who present with complex psychological and social difficulties. Relevant, up-to-date information should be made available to inform health professionals about the needs of current waves of refugees and asylum seekers and other available services for referral and collaboration. Health providers should ensure adequate time is allocated for appointments with refugees and asylum seekers allowing space for trust building, communication and cultural understanding. They should develop infrastructure to ensure that trained interpreters are provided in a timely manner for appointments. Where resources permit, trained interpreters should be available with face-to-face and remote options (eg, via phone), depending on patients' preferences.

### Policy

Healthcare policy makers and commissioners should recognise the complex needs of refugees and asylum seekers, providing enhanced resources for quality and equitable service provision. Adequate human resourcing would allow health professionals to spend the necessary time to follow best practice. Integration of specialised components with existing general practice may facilitate care. Asylum and resettlement policy makers should seek to promote continuity of relationship with healthcare providers, limiting relocations.

### Research

Primary qualitative research could explore other healthcare professionals' experiences of caring for refugees and asylum seekers. For example, no studies of pharmacists' experiences were identified in this review. Further systematic reviews could be conducted to investigate experiences of health professionals working with refugees and asylum seekers in other areas of the healthcare system. A systematic review of challenges and facilitators for mental health professionals providing services to refugees and asylum seekers could inform service delivery for this group and searches in for this current review identified primary studies that could be included.

The outputs from this review may be used to inform service models for refugees and asylum seekers.

Healthcare evaluations may be conducted to evaluate these models and identify areas that are able to improve quality of care.

### Strengths and limitations

An extensive and systematic search that was carried out across four databases complemented by reference and citation searches and it is therefore unlikely that published studies would have been overlooked. The inclusion of only English language studies may have led to under-representation of health professionals working in non-English speaking countries leading to a greater applicability to healthcare policy and practice in English-speaking high-income countries. It is also possible that the database searches may not have identified studies where refugees and asylum seekers were referred to as 'migrants' or 'immigrants'; however, the additional hand-searches conducted would likely have identified any further key studies relevant for this review.

In study selection, titles and abstracts were screened by one reviewer, giving potential for selection bias or for relevant studies to be missed. By involving a second reviewer at the full-text selection stage, the study team sought to minimise bias, and supplementary searches of reference lists and citations reduced the potential for missing key studies. A second reviewer in data extraction could have reduced possibility of transcription errors, and in the quality appraisal stage could have minimised potential for biased assessment. Ideally, the analysis process would also have involved multiple reviewers in coding and formation of descriptive and analytical themes, bringing a wider perspective to interpretation.

Participants in this review were limited to the core clinical professions of nurses, primary care doctors and midwives. Other professionals, that may be part of primary healthcare teams, such as mental health workers, counsellors, physiotherapists and other community workers, were not included, raising a question about the transferability to more diverse primary healthcare teams. Studies including other professional groups report similar themes to the present review; however, those including mental health professionals may have a greater emphasis on secondary stress experienced when working with traumatised patients.[85 86] A further consideration for transferability of these findings is the combining of data from the three clinical professions as they have different care practices, interaction with patients and support networks, giving the potential to introduce imprecision to the findings.

A strength of syntheses of qualitative research is that concepts are translated across studies, with common themes described that may be more transferable to other contexts and a greater ability to inform policy and practice.[26 87] This contrasts with primary qualitative studies that are tied to their context and transference of findings is treated with caution.[26 87] On the other hand, a perceived limitation of thematic syntheses is that they introduce a greater degree of abstraction from original experiences, sacrificing thickness of data and details found within the

primary studies.[88] In this case, given that refugees are not a homogeneous group, it is perhaps acceptable to emphasise only the more generalised themes that transcend the contexts of individual studies.

## CONCLUSIONS

Many people continue to be displaced due to conflict and persecution, seeking sanctuary in high-income countries. Health professionals that provide primary healthcare for refugees and asylum seekers experience a range of challenges and facilitators; within the healthcare encounter, the environment of the healthcare system, and in the broader context of asylum and resettlement policy and process. The challenges and facilitators identified in this review may inform practices and policies that improve the quality of healthcare and minimise healthcare inequalities for refugees and asylum seekers.

**Acknowledgements** The authors acknowledge the contribution of Madeleine Flawn who was a second reviewer in study selection stage of the review.

**Contributors** LR conceived the study and was responsible for the study design, search strategy, data extraction, quality assessment and data analysis. LLJ provided methodological advice. LR drafted the manuscript which was revised with input from LLJ and SD. All authors approved the final version for publication.

**Competing interests** None declared.

**Provenance and peer review** Not commissioned; externally peer reviewed.

**Data sharing statement** No additional data are available.

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
