## [Reviewer comments · BMJ Open]

ARTICLE DETAILS

TITLE (PROVISIONAL)	Challenges and Facilitators for Health Professionals Providing Primary Healthcare for Refugees and Asylum Seekers in High-Income Countries: A Systematic Review and Thematic Synthesis of Qualitative Research
AUTHORS	Robertshaw, Luke; Dhesi, Surindar; Jones, Laura

VERSION 1 - REVIEW

REVIEWER	Dr Margaret Kay The University of Queensland, Australia
REVIEW RETURNED	04-Feb-2017

GENERAL COMMENTS	Thank you for the opportunity to review this paper. I feel it is a valuable paper that provides useful insights into the complexities related to delivering refugee health care in the primary health care setting. This is a valuable paper that will assist health providers to engage with policy makers to improve the health landscape for health care delivery to vulnerable populations such as refugees and asylum seekers. The paper is easy to read and engages well with the existing literature on this topic. There are a few specific comments outlined below. Introduction The introduction provides an well-referenced background and rationale for the review Page 4 Line 15 – the apostrophe appears misplaced Methods: This section reads well, although it is unclear if new theme could be added if it arose within a paper that was analysed after the index paper. It would be useful to the reader to clarify this. Although the index paper was chosen because it was rich, it is hard to believe that it had every theme present in all 26 papers. Page 8 Line 39 It would appear unnecessary to add the words ‘ within an a priori framework’ of challenges and facilitators. It could be argued that this phrase unnecessarily absorbs a positivist framing. Adding this phrase offers no benefit to the paper. Results: Search and selection: Well presented Given the significant literature on refugee health from Canada and the strength of their primary health care system, it is a little surprising that there was only one paper and it is possible that the tendency to use the term migrant for refugee populations in some of the Canadian literature could have reduced the identification of relevant literature. I am not sure if the authors are able to comment on the possibility of this as a limitation to the search, given the results.
---

	Characteristics of studies: Well presented Table 1: Well presented and informative Detailed quality assessment provided was succinct and informative. Figure 2: Very clear Table 2: Well presented and informative The details of the results are clearly presented and relevant to the health care provider Discussion This is clearly presented and highlights the important issues and includes a focus on the public health issues relevant to policy, recognising the intersect with social determinants of health that are not necessarily articulated in the qualitative research itself. The recommendations for practice and policy are appropriate and useful. While the qualitative research does not capture the complexity of using interpreters, it is appropriate to point out that while advocating for face-to-face interpreters as a gold standard, there are times when the telephone interpreting is the necessary as this can maintain anonymity for the patient. This can be essential when the refugee community is relatively small and the interpreter will logically and unavoidably have community ties to the patient. Care should be taken when emphasising this recommendation. Policy recommendations could include adequate resourcing to enable providers to spend the necessary time suggested as best practice. Research recommendations could have included further research with other primary care practitioners such as pharmacists as research with these providers was identified as missing in the initial results. The Strengths and Limitations section presents a substantial discussion of these issues. Conclusions are relevant to the paper's aim and findings. References Some journal names are abbreviated and others are not – this would need to be consistent. References 1,2,3,4,5,7,18, 19,29,30, 56,57,75 seem incomplete – some seem to need a place of publication, others need a url link Reference 24: There is a 'box' rather than an initial for Fingeld in this reference References 35, 51, 54, 58,60,61,62, 63, 65-74, 76, 78, 81-83: There is a comma after the title rather than a fullstop Reference 36: ends with p which seems odd Reference 40: does not need May-Jun included Reference 42: needs correction of its errors Reference 53: needs updating as it is now in print and there is a comma after the title rather than a fullstop Reference 77: The suggested citation for this report is available in the report and should be followed.
--	---

REVIEWER	Professor Kate O'Donnell General Practice & Primary Care Institute of Health & Wellbeing University of Glasgow Scotland
REVIEW RETURNED	23-Feb-2017

GENERAL COMMENTS	Robertshaw et al. Challenges and facilitators for health professionals providing primary
---

healthcare for refugees and asylum seekers in high-income countries: A systematic review and thematic synthesis of qualitative research.

This is an interesting and timely paper, with a systematic review in this area certainly needed. I enjoyed reading the paper and know it will add to the literature on health care for refugees and asylum seekers. I have a few comments which I hope will aid the clarity and reporting of the paper.

1. Introduction: The authors, rightly, identify infectious diseases, mental health and gender based violence as areas of health concern with this population. It would also be helpful, however, to acknowledge that there is an increasing recognition that refugees and asylum seekers also suffer from NCDs, such as CVD and diabetes, which can also be exacerbated when they have to flee their country of origin.

2. Introduction, Line 45: Could the authors expand on what they mean when they write about 'challenges that may contribute to recognised healthcare inequalities'. There are references which address this, in addition to the one cited. For example the paper by Ingleby Psychosocial Intervention 2012; 21: 331 or the more recent paper by O'Donnell et al Health Policy 2016; 120: 495 both discuss health systems as a structural determinant of health.

3. Introduction: I think the authors could be clearer about their focus on high-income countries. Why only high income countries, when we know that it is LMICs that bear the biggest burden in times of offering a home to refugees and asylum seekers. I assume this might be related to their interest in primary health care systems, but this needs to be articulated more clearly.

4. Methods, Page 6, Line 34: I am unfamiliar with the SPIDER tool. A little more clarification on why it was selected and how it was used would be useful.

5. Methods: It is clear that, at several points, only one member of the team was involved in screening and data extraction. This is a weakness of the study – as acknowledged in the limitations. Could the team reflect on whether they think anything pertinent was missed in this process. In addition, where this is detailed in the methods, could they indicate which author(s) was involved.

6. Methods, Page 7, Line 44-45: The authors state that papers focused on 'illegal immigrants, undocumented migrants, migrants or immigrants' were excluded. I would like to see clearer justification for this, especially in relation to the exclusion of papers dealing with illegal or undocumented migrants. For example, this might be discussed in the final discussion.

7. Data synthesis, Page 8: I am unclear why the team took the approach of starting with a single 'data rich' paper. Generally, the process of developing a coding framework might be based on several papers. It would be beneficial, therefore, to know more about how the first paper was selected and how much additional coding was derived from later papers. There is also a tension between coding inductively and having an a priori framework – how was this tension managed. Finally, how was the a priori framework arrived at?

8. In the presentation of the results, it would be helpful to know if the themes identified were spread evenly across the papers or if some were more characteristic of a particular health care system or asylum system than others. This might be particularly important when considering the themes in relation to health systems or to asylum and resettlement.

	Overall, this is an interesting paper and the systematic identification and collation of references will make it a very useful paper in the field.
--	--

VERSION 1 – AUTHOR RESPONSE

Reviewer 1: Dr Margaret Kay

Comment	Amendment
Introduction	
Page 4 Line 15 – the apostrophe appears misplaced	Many thanks, this has now been corrected.
Methods	
This section reads well, although it is unclear if new theme could be added if it arose within a paper that was analysed after the index paper. It would be useful to the reader to clarify this. Although the index paper was chosen because it was rich, it is hard to believe that it had every theme present in all 26 papers.	We thank the reviewer for highlighting the lack of clarity about the ability for all papers to contribute to themes. The themes were developed using the final codebook and all papers and all papers had the potential to add new codes. Starting with a paper known to contain numerous challenges and facilitators was a pragmatic decision to help the researcher develop a broad range of codes early in the process. The following has been added to the Methods section to bring further clarity: Concepts in each article were coded to iteratively develop and refine a codebook, with each article having an ability to contribute new codes. Once all articles had been coded, the finalised codebook was applied across all articles.
Page 8 Line 39 It would appear unnecessary to add the words ‘within an a priori framework’ of challenges and facilitators. It could be argued that this phrase unnecessarily absorbs a positivist framing. Adding this phrase offers no benefit to the paper.	Thank you for your comments about the phrase ‘within an a priori framework’. After discussion with the research team, would like to make it clear that the research question provided two pre-defined categories (challenges and facilitators) that set a framework for coding. Within these categories, an inductive approach was taken. The following sentences have been modified/added to bring further clarity to this point: The findings (results) and discussion sections were coded inductively within the two categories of ‘challenges’ and ‘facilitators’, as defined by the review question. This approach of inductive coding

Comment	Amendment
	within a priori categories follows established methodology seen in similar qualitative syntheses.[36]
Results	
Given the significant literature on refugee health from Canada and the strength of their primary health care system, it is a little surprising that there was only one paper and it is possible that the tendency to use the term migrant for refugee populations in some of the Canadian literature could have reduced the identification of relevant literature. I am not sure if the authors are able to comment on the possibility of this as a limitation to the search, given the results.	We thank the reviewer for raising the possibility of Canadian articles being missed in the search strategy because of the tendency to use ‘migrant’ for refugee populations. It is agreed that ‘migrant’ could have been included in the search strategy to capture any such cases; however, it would have significantly widened the scope of the search as migrant includes a wider population such as economic migrants, students and immigrants in general. Because of this, a pragmatic decision was made to limit the search to those described as refugees and asylum seekers. The additional hand searches of references and citations would likely have identified any other key articles within the literature. The following has been added to the strengths and limitations section: It is also possible that the database searches may not have identified studies where refugees and asylum seekers were referred to as ‘migrants’ or ‘immigrants’; however, the additional hand-searches conducted would likely have identified any further key studies relevant for this review.
Discussion	
While the qualitative research does not capture the complexity of using interpreters, it is appropriate to point out that while advocating for face-to-face interpreters as a gold standard, there are times when the telephone interpreting is the necessary as this can maintain anonymity for the patient. This can be essential when the refugee community is relatively small and the interpreter will logically and unavoidably have community ties to the patient. Care should be taken when emphasising this recommendation.	We agree that the original wording was directional and have modified the recommendation as follows: Where resources permit, trained interpreters should be available with face-to-face and remote options (e.g. via phone), depending on patients’ preferences.
Policy recommendations could include adequate resourcing to enable providers to spend the necessary time suggested as best practice.	We would like to thank the reviewer for this suggestion. The following sentence has been added to the policy recommendation section:

Comment	Amendment
	Adequate human resourcing would allow health professionals to spend the necessary time to follow best practice.
Research recommendations could have included further research with other primary care practitioners such as pharmacists as research with these providers was identified as missing in the initial results.	This is a helpful point and we have included the following research recommendation: Primary qualitative research could explore other healthcare professionals' experiences of caring for refugees and asylum seekers. For example, no studies of pharmacists' experiences were identified in this review.
References	
Some journal names are abbreviated and others are not – this would need to be consistent.	Many thanks, this has now been corrected
References 1,2,3,4,5,7,18, 19,29,30, 56,57,75 seem incomplete – some seem to need a place of publication, others need a url link	Many thanks, this has now been corrected
Reference 24: There is a 'box' rather than an initial for Fingeld in this reference	Many thanks, this has now been corrected
References 35, 51, 54, 58,60,61,62, 63, 65-74, 76, 78, 81-83: There is a comma after the title rather than a full stop	Many thanks, this has now been corrected
Reference 36: ends with p which seems odd	Many thanks, this has now been corrected
Reference 40: does not need May-Jun included	Many thanks, this has now been corrected
Reference 42: needs correction of its errors	Many thanks, this has now been corrected
Reference 53: needs updating as it is now in print and there is a comma after the title rather than a fullstop	Many thanks, this has now been corrected
Reference 77: The suggested citation for this report is available in the report and should be followed.	Many thanks, this has now been corrected

Reviewer 2: Professor Kate O'Donnell

Comment	Amendment
Introduction	
The authors, rightly, identify infectious diseases, mental health and gender based violence as areas of health concern with this population. It would also be helpful, however, to acknowledge that there is an increasing recognition that refugees and asylum seekers also suffer from NCDs, such as CVD and diabetes, which can also be exacerbated when they have to flee their country of origin.	Many thanks to the reviewer for this valuable insight. We have added the following sentence to the introduction: Refugees and asylum seekers also suffer from non-communicable diseases such as hypertension, musculoskeletal disease, chronic respiratory disease and diabetes, which may be under-managed and exacerbated when they are forced to flee their countries.[10]
Line 45: Could the authors expand on what they mean when they write about 'challenges that may contribute to recognised healthcare inequalities'. There are references which address this, in addition to the one cited. For example the paper by Ingleby Psychosocial Intervention 2012; 21: 331 or the more recent paper by O'Donnell et al Health Policy 2016; 120: 495 both discuss health systems as a structural determinant of health.	Thank you for providing further references that may strengthen the introduction to healthcare inequalities. This paragraph has been reworked to include health systems as a determinant in health inequality and the reference to O'Donnell et al 2016 is now included. Primary healthcare teams are on the front-line of healthcare provision for refugees and asylum seekers that arrive in high-income countries.[14] These teams may include a variety of professional backgrounds, clinical and non-clinical, but typically include a core of general practitioners, community based nurses and midwives.[15, 16] These health professionals face significant challenges when caring for refugees and asylum seekers.[17-19] They must address their complex health and social needs, often in cross-cultural interactions, and operate within health systems that may not be structurally configured or politically favourable towards this group.[17-20] These challenges impact on their ability to provide the same quality of care as the general population, leading to healthcare inequalities.[20, 21]
I think the authors could be clearer about their focus on high-income countries. Why only high income countries, when we know that it is LMICs that bear the biggest burden in times of offering a home to refugees and asylum seekers. I assume this might be related to their interest in primary health care systems, but this	We agree that a clearer statement about the focus on high income countries is needed. The following sentence has been added to the methods section: Studies from high-income countries were selected because the authors were interested in the development of recommendations for policy and

Comment	Amendment
needs to be articulated more clearly.	practice applicable in advanced primary healthcare systems.
Methods	
Page 6, Line 34: I am unfamiliar with the SPIDER tool. A little more clarification on why it was selected and how it was used would be useful.	We agree that this requires further clarification. This sentence has been expanded as follows: The search strategy was based on the SPIDER (Sample, Phenomenon of interest, Design, Evaluation, Research type) tool, which has been developed as an alternative to PICO (Population, Intervention, Comparison, Outcome) to optimise identification of qualitative studies for evidence syntheses.[30]
It is clear that, at several points, only one member of the team was involved in screening and data extraction. This is a weakness of the study – as acknowledged in the limitations. Could the team reflect on whether they think anything pertinent was missed in this process. In addition, where this is detailed in the methods, could they indicate which author(s) was involved.	As recommended, further reflection on this issue has been added to strengths and limitations as follows: In study selection, titles and abstracts were screened by one reviewer, giving potential for selection bias or for relevant studies to be missed. By involving a second reviewer at the full-text selection stage, the study team sought to minimise bias, and supplementary searches of reference lists and citations reduced the potential for missing key studies. Also, the Initials (LR) added in three places (Screening, data extraction, quality assessment) in the methods to help with clarity.
Page 7, Line 44-45: The authors state that papers focused on ‘illegal immigrants, undocumented migrants, migrants or immigrants’ were excluded. I would like to see clearer justification for this, especially in relation to the exclusion of papers dealing with illegal or undocumented migrants. For example, this might be discussed in the final discussion.	Thank you for raising the need for justification of the selection criteria used. It was felt that this may be best placed in the methods section where other justifications are given for selection criteria. Therefore, the methods section has been expanded to include: Articles that had referred to service users as ‘migrants’ or ‘immigrants’ were excluded, as these terms have a broader meaning including economic migrants, students and family unification.[32] Those that referred to ‘illegal immigrants’ or

Comment	Amendment
	'undocumented migrants' were also excluded as they are known to have unique characteristics (e.g ineligible for free healthcare) that would not be typical of refugees and asylum seekers.[33]
Data synthesis, Page 8: I am unclear why the team took the approach of starting with a single 'data rich' paper. Generally, the process of developing a coding framework might be based on several papers. It would be beneficial, therefore, to know more about how the first paper was selected and how much additional coding was derived from later papers. There is also a tension between coding inductively and having an a priori framework – how was this tension managed. Finally, how was the a priori framework arrived at?	Thank you for highlighting the lack of clarity with the coding process. We have sought to clarify the coding process and developing the codebook. All papers could contribute new codes and influence the coding framework. Concepts in each article were coded to iteratively develop and refine a codebook, with each article having an ability to contribute new codes. Once all articles had been coded, the finalised codebook was applied across all articles. Regarding the a priori framework, the description has been expanded: The findings (results) and discussion sections were coded inductively within the two categories of 'challenges' and 'facilitators', as defined by the review question. This approach of inductive coding within a priori categories follows established methodology seen in similar qualitative syntheses.[34]
Results	
In the presentation of the results, it would be helpful to know if the themes identified were spread evenly across the papers or if some were more characteristic of a particular health care system or asylum system than others. This might be particularly important when considering the themes in relation to health systems or to asylum and resettlement.	Thank you for this helpful suggestion. It is agreed that an analysis of the contribution of particular health systems to the themes would be an interesting development of the study and provide more information for interpretation. As this further analysis was not part of the objectives for this review, it has been considered beyond the scope of the project by the research team. This suggestion will certainly be a useful consideration when planning future thematic syntheses.

VERSION 2 – REVIEW

REVIEWER	Dr Margaret Kay The University of Queensland, Australia
REVIEW RETURNED	14-Apr-2017

GENERAL COMMENTS	I have enjoyed this paper and feel that the authors' responses and revisions were carefully considered the alterations have improved this paper significantly. There is great value in having this manuscript published and I recommend that it be accepted for publication. I would like to thank the authors for the opportunity to review this excellent paper.
---

REVIEWER	Professor Kate O'Donnell General Practice & Primary Care, Institute of Health & Wellbeing, University of Glasgow
REVIEW RETURNED	18-Apr-2017

GENERAL COMMENTS	I am now happy to recommend for publication.
--